# Solution-Mediated Inversion of SnSe to Sb_2_Se_3_ Thin-Films

**DOI:** 10.3390/nano12172898

**Published:** 2022-08-23

**Authors:** Svetlana Polivtseva, Julia Kois, Tatiana Kruzhilina, Reelika Kaupmees, Mihhail Klopov, Palanivel Molaiyan, Heleen van Gog, Marijn A. van Huis, Olga Volobujeva

**Affiliations:** 1Department of Materials and Environmental Technology, School of Engineering, TalTech, Ehitajate tee 5, 19086 Tallinn, Estonia; 2Auramet Solutions OÜ, Kalliomäentie 1B, 02920 Espoo, Finland; 3Department of Cybernetics, School of Science, TalTech, Ehitajate tee 5, 19086 Tallinn, Estonia; 4Research Unit of Sustainable Chemistry, Faculty of Technology, University of Oulu, Pentti Kaiteran katu 1, 90014 Oulu, Finland; 5Nanostructured Materials and Interfaces, Zernike Institute for Advanced Materials, University of Groningen, Nijenborgh 4, 9747 AG Groningen, The Netherlands; 6Soft Condensed Matter, Debye Institute for Nanomaterials Science, Utrecht University, Princetonplein 5, 3584 CC Utrecht, The Netherlands

**Keywords:** chemical transformation, ion exchange, doping, thin films, DFT calculation

## Abstract

New facile and controllable approaches to fabricating metal chalcogenide thin films with adjustable properties can significantly expand the scope of these materials in numerous optoelectronic and photovoltaic devices. Most traditional and especially wet-chemical synthetic pathways suffer from a sluggish ability to regulate the composition and have difficulty achieving the high-quality structural properties of the sought-after metal chalcogenides, especially at large 2D length scales. In this effort, and for the first time, we illustrated the fast and complete inversion of continuous SnSe thin-films to Sb_2_Se_3_ using a scalable top-down ion-exchange approach. Processing in dense solution systems yielded the formation of Sb_2_Se_3_ films with favorable structural characteristics, while oxide phases, which are typically present in most Sb_2_Se_3_ films regardless of the synthetic protocols used, were eliminated. Density functional theory (DFT) calculations performed on intermediate phases show strong relaxations of the atomic lattice due to the presence of substitutional and vacancy defects, which likely enhances the mobility of cationic species during cation exchange. Our concept can be applied to customize the properties of other metal chalcogenides or manufacture layered structures.

## 1. Introduction

The global tendency towards carbon neutrality puts high demands on the elaboration of future applications. The role of novel materials in energy transition already requires significant technological advancements because of the toxicity of active elements and/or their scarcity. In such conditions, reaching advances in developing suitable technologies and materials becomes crucial for the sustainable growth of the green economy.

Converting one substance into another by partial or complete ion exchange (IE) can be a powerful tool for producing materials with specified or new properties [1,2,3,4,5,6,7,8,9,10,11,12]. For half a century, significant progress has been achieved in the ion exchange synthesis of numerous nanoscale materials with structural and compositional diversity. In contrast, minimal success has been demonstrated in the inversion of extended solid materials attached to substrates, which significantly limits the applicability of ion exchange as a synthetic route [9]. Vacancy-assisted cation diffusion and kick-off mechanisms have been suggested to control ion replacement [1,4]. Implying either of the two proposed IE reaction paths, the conversion of bulk materials may be impractical due to the necessity for high temperatures or pressures to overcome the activation barrier for the diffusion of constituents. However, it should be noted that numerous properties of extended solids and characteristics of reaction systems are not sufficiently considered for a correct assessment of IE thermodynamics and kinetics. For instance, the concentration of defects, the contribution of grain boundaries (GBs), and the presence of impurities acting as catalysts can significantly reduce the temperature for material transformation. Further decrease in the crystallite size and increase in amorphous-to-crystalline phase ratio could mimic the kinetics and mechanisms inherent to nanomaterials, yielding the synthesis of kinetically-controlled micro- and macromaterials [13]. Comprehending the key principles underlying the scale-dependent chemical transition in bulk is required when designing new chemical routes to expand the spectrum of macro-, micro-, and even nanoscale materials. 

Herein, we demonstrate a solution-assisted partial and complete aliovalent cation substitution that develops on the scale of thin films with an unexpectedly high reaction rate. We show that the SnSe template interacts with a source of Sb^3+^ cations to create high-quality Sb_2_Se_3_ layers, an alternative absorber material for light-harvesting applications [14,15] and next-generation lithium (sodium)-ion battery anodes [16]. The direct transformation to Sb_2_Se_3_ is awe-inspiring in the context of unfavorable thermodynamics for antimony ion-assisted exchange reactions [17]. However, some defects with excessive energy and their specific concentration in SnSe can generate conditions for high ion mobility. This feature of extended solids can be modulated to facilitate complete conversion and become sufficient to minimize the impact of thermodynamic constraints and attenuated kinetics at relatively low temperatures. 

## 2. Results and Discussion

### 2.1. Discussion on the Stability of Antimony Selenide (Sulfide)

The internal stabilities of photoactive materials, their compatibilities, and their resistance to certain environmental conditions determine light-assisted devices’ efficiency and operational stability. Despite the general assumption that metal sulfide/selenide are stable, the fabrication of phase-pure constituent layers is challenging compared to silicon materials [14,18,19]. A relatively soft lattice, the presence of unstable defects, and a narrow *Eh* stability window regardless of pH to be employed (Figure 1a,b, Pourbaix diagram) trigger substantial changes in antimony selenide/sulfide and similar metastable semiconducting compounds under illumination, including oxide/hydroxide groups incorporation, photochemical reactions such as light-induced decomposition, surface passivation during wet-chemical synthesis, postdeposition treatment, and/or later during storage [19]. Although there is no precise theory clarifying the origin of photo-induced changes in metal chalcogenides, our recent study demonstrates the phase segregation and decomposition of quaternary, ternary, and binary compounds accelerated under light and humidity conditions [19]. In addition, the charge carrier transfer dynamics can depend on the facet orientation [14,18,20]. Relevant structure defects can influence charge carrier transport, accumulating excessive light-induced carriers within antimony sulfide/selenide materials to accelerate their degradation (Figure 1c). Therefore, certain synthesis-derived deviations in the phase composition and purity, structural properties, and structure of defects in antimony chalcogenides that may occur during fabrication under illumination are essential aspects to consider for material stabilization. Thus, fabricating single-phase antimony chalcogenides and similar compounds with enhanced stability is still a challenge and requires additional efforts to promote the development of high-performance and durable devices.

### 2.2. Ion Exchange from SnSe to Sb_2_Se_3_

#### 2.2.1. Preliminary Trials

Thermal treatments and storage in an oxygen/moisture-containing atmosphere convert photoactive antimony sulfide/selenide to antimony (III) oxide/hydroxide because of their internal instability (Figure 1a–c). The inclusion of oxide/hydroxide groups makes the planes (hk0) dominant over (hk1) in bulk Sb_2_S(Se)_3_, impairing the device performance [14,21]. Precise control over the incorporation kinetics of O/OH groups during the synthesis or postdeposition treatment can assumedly enhance the material stability and power conversion efficiency (PCE) of light-assisted devices. Our recent report demonstrated selective elimination of antimony-rich oxyselenide from sputtered Sb_2_Se_3_ films via partial postdeposition IE leading to enhanced device efficiency [14]. We implemented the main features of the proposed technology to create high-quality Sb_2_Se_3_ thin films by allowing Sb^3+^ cations to replace Sn positions in more stable SnSe template layers. 

Despite the apparent IE simplicity and idea straightforwardness, numerous technical obstacles such as emerging cracks, appearing undesired holes, and weak adhesion of the forming material to the substrate often occur. To overcome these limitations, we have varied the concentration of SbCl_3_, reaction time, deposition temperature for template layers, and partly types of substrates to minimize film-to-substrate lattice mismatch for better controlling the quality of Sb_2_Se_3_ films. From a substrate type point of view, sodium glass substrates require more gentle conditions for ion exchange as films can easily peel off from the substrate during SnSe-to-Sb_2_Se_3_ transformation. Substrates such as Mo, FTO, and ITO can withstand more aggressive exposure conditions and lead to the formation of continuous films and structures. Deposition of SnSe layers at temperatures exceeding 300 °C with a high crystallinity degree is preferable for forming materials well-attached to the substrate upon IE. Templating at temperatures below 250 °C stabilizes a high concentration of defects and yields instability during ion replacement. Concentrations of SbCl_3_ in glycerol exceeding 45 mM destroy the adhesion of the forming layer regardless of the substrate used. When critical limits preventing successful IE were determined, we studied the effect of SbCl_3_ concentrations and treatment time for IE to reach the desired quality of Sb_2_Se_3_ layers.

#### 2.2.2. Concentration of SbCl_3_

The scanning electron microscope (SEM) images demonstrate the remaining unchanged thickness of about 700 nm for all formed layers, indicating the absence of corrosive processes (Figure 2a). This immutability of the film thickness regardless of antimony salt concentrations used or treatment time demonstrates a fundamental characteristic of cation exchange realized in the number of anions on the substrate remaining unchanged during the exchange reaction [5]. Multiple data analysis of the film treated in 11 mM SbCl_3_ solution for 22 min reveals the formation of a layered structure with a crystalline top belonging to heavily antimony-doped orthorhombic SnSe (Figure 2, Appendix A) and a bottom porous mostly amorphous slightly antimony-doped SnSe layer attached to the substrate (Appendix A). The chemical compositions of 22, 33, and 44 mM-treated films were confirmed by the energy-dispersive X-ray (EDX) analysis, which shows strong signals of Sb and Se only with atomic ratios of ~2:3 (Appendix A), consistent with the stoichiometry of Sb_2_Se_3_. Thus, concentrations of SbCl_3_ exceeding 11 mM induce the complete transition of SnSe templates to Sb_2_Se_3_ (Figure 2a and Appendix A). 

In order to disclose the structural characteristics caused by cation exchange, the films were investigated by X-ray diffraction (XRD) and Raman spectroscopy (Figure 2b,c, Appendix A). Briefly, Sb_2_Se_3_ samples crystallize in orthorhombic symmetry with the Pbmn space group. All the diffraction peaks of films after conversion are properly consistent with the pattern of the Sb_2_Se_3_ single crystal in PDF card No.: 01-089-0821, revealing the high phase purity of fully converted samples. It is important to note that the main recorded XRD reflections coincide with the (101), (211), (221), (301), (311), and (151) planes. These structural (hk1) features relating to vertically aligned (Sb_4_Se_6_)_n_ ribbons are necessary for proper control of charge carrier transport in the absorber layer [21,22]. The suppressed growth of Sb_2_Se_3_ crystallites along the (hk0) planes with laterally-oriented (Sb_4_Se_6_)_n_ ribbons confirms the kinetically-driven cation replacement. The Raman spectrum of the pristine SnSe shows five distinguishable peaks positioned at 70.4, 97.1, 119.5, 156.0, and 184.0 cm^−1^ (black curve in Figure 2c, Appendix A), which differ from those at around 98.7, 116.0, 129.0, 153.3, 190.8, and 211.9 cm^−1^ (royal blue curve in Figure 3c, Appendix A) recorded for the 22 mM-processed films being completely transformed to Sb_2_Se_3_. A notable redshift in the relative peak positions observed for the 11 mM treated sample as compared to the pristine SnSe relates to the development of internal strain due to the Sb incorporation into the lattice [14]. 

#### 2.2.3. Time Treatment

The reaction speed in semicrystalline template layers attached to the substrate at relatively low temperatures is amazing (Figure 3). In a separate attempt with SnSe thin films deposited at temperatures exceeding 500 °C, we noticed that cation replacement is practically hindered under similar experimental conditions. For accelerated cation replacement in extended solids, numerous materials may require direct contact with molten salts at high temperatures [17,23,24,25]. The reaction time is significantly reduced for systems with a high defect concentration than can be determined in crystalline systems with a low defect concentration. This fact indicates that the activation barrier for ion replacement can be adjusted by playing with various concentrations of defects in bulk, shifting the molecule-like dynamic equilibrium towards the desired reaction products. We present a schematic way to fabricate attached-to-substrate thin-film materials using a cation replacement in Figure 4.

The initial handling of the sample in the 44 mM SbCl_3_ solution for 5 min causes a twofold increase in the crystallite size of SnSe (Appendix A). This circumstance, combined with changes in the lattice parameters to the bulk values, unaltered the phase and elemental compositions as confirmed by XRD and EDX (Appendix A) and remained the same Raman peak positions recorded for the pristine and 5 min treated films (Appendix A), indicates the absence of internal stress due to an inactivated substitution of Sn ions. In contrast, the processing from 6 to 9 min at ~210 °C induces a monotonous diminution in the lattice parameters along with the a and b axes, as confirmed by a notable shift of the (111) XRD peak located at ~30.7° towards higher 2*θ* values (Figure 3b,c). This fact, accompanied by the unchanged elemental composition of the 6 min film (Appendix A) and an alteration in Raman peak positions to lower wavenumbers (Appendix A) due to the tensile strain evolution, might indicate the subsequent formation of tin vacancies driven by interaction with the multi-ion system [1]. Effective compositional tailoring that involves the replacement of Sn^II^ ions with radii (118 pm) [26] bigger than Sb^III^ (90 pm) [27] into lattice sites launches after 7 min. A slightly longer process of 10 min yields a significant increase in the Sb content up to 7.9% (Appendix A), with higher Sb concentrations on the film surface and lower concentrations close to the Mo substrate, similarly to those shown in Appendix A. Ion exchange lasting 12 min promotes further inclusion of Sb up to ~11.4 % (12-min sample in Figure 3a), creating conditions for a notable phase transition. Several peaks located at, e.g., 25.9°, 29.0°, 30.3°, and 44.1° (magenta curve in Figure 3b) deviate from those observed in the XRD patterns of SnSe and Sb_2_Se_3_ films synthesized within this study, hinting at an intermediate crystal structure. Raman studies support the data obtained from XRD and SEM/EDX measurements on forming a metastable ternary compound from a Sn_x_Sb_y_Se_z_ family, as the Raman spectrum of the 12 min handled sample shows bands, which are entirely different from those of SnSe and Sb_2_Se_3_ (Appendix A). Seventeen and twenty-two min processed samples are fully transformed to Sb_2_Se_3_, as revealed from the EDX data showing the disappearance of Sn signal and the XRD patterns and Raman spectra being completely dissimilar from the ground state and the 12 min exposed sample (Figure 3b and Appendix A). Furthermore, a short processing time (≤10 min) yields the material amorphization, while an extended processing time launches a significant increase in crystallite size after the matrix is turned into Sb_2_Se_3_ (Appendix A). 

#### 2.2.4. Discussion on the Ion-Exchange Mechanism

Despite decades of research on various ion-exchange reactions, the mechanism for Sb substitution remains a mystery [14,17]. A moderate amorphization or high degree of disorder during the Sn-to-Sb replacement observed in our experiments, along with the data discussed above, reveals that Sb predominantly accumulates at grain boundaries or interfaces. Thus, a certain concentration of GBs or surface/interface imperfections may facilitate ion exchange reactions in bulk materials due to numerous defect states originating from the structural disorder [28,29]. The electronic structure of two-dimensional extended defects in SnSe is dominated by anion–anion (Se_i_−Se_i+1_) and/or cation–cation (Sn_i_−Sn_i+1_) wrong bonds [30]. Possible replacements of 4p−4p anion-core dangling bonds with Se−Sb with the higher energy of 4p−5p states are energetically unfavorable to initiate ion exchange at the earliest transition stages. Structural and compositional changes observed in 5-to-8 min treated samples using XRD, Raman, and SEM/EDX measurements suggest that Sn−Sn bonds might be the initial centers for starting cation exchange at interfacial defects, as expected from tin-rich conditions (Appendix A). Transition starts with the passivation of strained Sn−Sn bonds (5p−5p) by hydrogen and later chlorine ions with the formation of Sn−Cl units with energetically favored 5p−3p states (Appendix A), a quite common observation for different optoelectronic materials [14,18,31]. Numerous species presented in solution provoke the generation of tin vacancy defects (Appendix A) at the interface planes with subsequent extraction of tin atoms driven by complexation with chlorine ions and various glycerin forms [32]. High concentrations of cationic vacancies (V_Sn_) facilitate the penetration of highly charged Sb^3+^ into the defective material and the occupation of available V_Sn_ (Appendix A) to ensure electrical neutrality. Experimental data suggest that Sn-to-Sb cation replacement passes through three main stages: (i) incorporation of Sb to SnSe to approximately 8 at. %; (ii) relative transition point with an approximate composition Sn_37_Sb_11_Se_52_ (idealized to Sn_3_Sb_1_Se_4_); and (iii) formation of phase-pure Sb_2_Se_3_ after continued cation exchange (Appendix A).

We conducted DFT calculations to estimate the theoretical capabilities of SnSe to adopt Sb to the level when the formation of possible intermediate phase and subsequent complete phase transition to Sb_2_Se_3_ becomes thermodynamically favorable. Computational details can be found in the Methods section, and an overview of the relevant phases and their calculated lattice parameters can be found in Appendix A. To assess the thermodynamic stability of the Sb content in SnSe, the number of atoms of each species needs to be equal on both sides of the reaction. To complete the energy balance, we corrected the changing cation/anion ratio (Me/Se) by compensating with the energy of *β*−*Sn*^0^, which is expelled when the Me/Se ratio of compounds shifts from 1.0 to 0.667 and is expected under experimental tin-rich conditions. So, e.g., for a compound Sn_2_Sb_2_Se_4_, the reaction formula reads:(1)$$\left(Sn_{2}Sb_{2}\right)Se_{4}\to Sb_{2}Se_{3}+SnSe+Sn$$

Equation (1) can be generalized as follows for an impurity cationic concentration *x* of Sb within SnSe:(2)$$Sn_{\left(1-x\right)}Sb_{x}Se\to \;\frac{x}{2}\;\left\{Sb_{2}Se_{3}\right\}\;+\;(1-\frac{3}{2}x)\cdot \{\mathrm{SnSe}\}\;+\;\frac{x}{2}\;\mathrm{Sn}$$

We evaluated the formation enthalpy of Sb-doped SnSe with respect to the reference energies of defect-free Sb_2_Se_3_, SnSe, and Sn phases using the relation:(3)$$∆H_{1\;}=E\;\{Sn_{\left(1-x\right)}Sb_{x}Se\}-\frac{x}{2}\;\cdot E\{{\mathrm{Sb}}_{2}{\mathrm{Se}}_{3}\}-(1-\frac{3}{2}x\;)\;\cdot E\{\mathrm{SnSe}\}\;-\frac{x}{2}\;E\{\mathrm{Sn}\}$$
where all energies are in eV/atom. For varying the antimony content, two supercells of SnSe such as 2 × 2 × 1 and 1 × 3 × 1 (containing 16 Sn and 16 Se atoms, and 12 Sn and 12 Se atoms, respectively) were constructed with periodic boundary conditions in all directions. The number of Sb atoms substituting Sn atoms in the supercell was then systematically increased to cover the energetics of Sb-for-Sn substitution in the range 0 < *x*_Sb_ < 0.4 (atomic fractions). The formation enthalpies calculated using Equation (3) are listed in Appendix A. Figure 5 displays the formation enthalpy of these phases as a function of Sb concentration. The negative slope (d*H*_1_/d*x_Sb_* < 0) of the curve (Figure 5) confirms the energetically preferred occupation of Sb of the cationic positions in SnSe and refutes the splitting Sn_1-x_Sb_x_Se into Sn, SnSe, and Sb_2_Se_3_ if the Sb atomic fraction is below 0.4. Therefore, when cation replacement is halted while the phase is still SnSe-type, the Sb-doped SnSe structures are thermodynamically stable or at least metastable. In the literature, the ternary Sn−Sb−Se phase diagram has not been explored in detail. Ismailova et al. experimentally studied the phase equilibria in the SnSe−Sb_2_Se_3_−Se system [33]. The compositions Sn_1-x_Sb_x_Se investigated in the present work with a cation/anion ratio of unity fall outside the reported compositional triangle, [33] though. The exact crystal structures of the Sn_2_Sb_2_Se_5_ and SnSb_2_Se_4_ ternary phases mentioned by Ismailova et al. are unknown and therefore were excluded from the DFT calculations. Their lattice parameters are so large (more than 2 nm along two crystal axes), however, that crystallization of such large unit cells is very unlikely to occur during the fast and intrusive cation exchange process studied here.

The relative thermodynamic stability of the ion-exchanged Sn_1−x_Sb_x_Se phases, as predicted in the current study, can be overcome by applying a strong chemical potential to the Sn atoms, which experimentally are found to be extracted from the thin films. For example, when not elemental Sn but solid SnCl_2_ is considered as a reference phase for the formation enthalpy, our DFT calculations predict that the chemical potential would shift by −3.15 eV per Sn atom. By taking this chemical potential shift into account, a formation enthalpy Δ*H*_2_ can be defined corresponding to the following equation:(4)$$∆H_{2}=∆H_{1}-\frac{x}{2}∆\mu_{Sn}=∆H_{1}-\frac{x}{2}\left(3E\left\{SnCl_{2}\right\}-E\left\{Sn\right\}-2E\left\{Cl_{2}\right\}\right)\;$$
where *μ*_Sn_ is the chemical potential of Sn, and all energies are expressed in eV/atom. The formation enthalpies Δ*H*_2_ calculated in this way are also plotted, as the red curve in Figure 5, and this chemical potential shift renders the formation enthalpies of the Sn_1−x_Sb_x_Se phases nearly zero (higher than −0.07 eV/atom), so that phase separation yielding pure Sb_2_Se_3_ is more likely to occur. DFT calculations cannot calculate the energy of the relevant ions in the CE solution, though, which determines the real chemical potentials during cation exchange.

The formation enthalpies listed in Appendix A and plotted in Figure 5 provide information about the relative phase stability of partly ion-exchanged SnSbSe phases. A full description of the dynamics of the cation exchange process, however, also requires information about migration energies of cationic species and possible interactions with vacancies, which was not included in the present study. Furthermore, to form the Sb_2_Se_3_ type of structure from the SnSe type of structure, the rectangular building blocks of the SnSe structure (displayed in Appendix A on the left-hand side) need to shift into corner-sharing configurations (displayed in Appendix A on the right-hand side) and the dynamics of this transformation would need to be simulated as well. Such investigations could be the topic of a more extensive follow-up study employing dynamical simulations (e.g., using force-field molecular dynamics).

The chemical solutions used in this study are multifunctional and demonstrate not only a simple transport role in delivering Sb ions into the reaction space. Various chemical species such as H^+^, Cl ions, and different glycerin forms presented in solutions probably assist in extracting excessive metallic inclusions from metal chalcogenide layers [14] to generate enough concentration of cationic vacancies, strongly shifting the chemical potentials and promoting the transformation of Sb-doped SnSe structures to Sb_2_Se_3_ (Appendix A). A specific mechanism of cation extraction at the current stage is unclear and requires additional studies of the solution and gaseous phases to clarify the processes occurring in the demonstrated cation exchange systems. Nevertheless, the final concentrations of Sn ions (~0.1–0.3 mM) formed in glycerol solutions are significantly lower than concentrations of Sb sources (11–44 mM). Such a high difference in the concentrations of competing cations (Sn vs. Sb) will encourage pulling Sn cations out of the film and prevent reverse cation exchange.

Preferential formation of (hk1) crystallographic planes indicates the anisotropic nature of the cation replacement realized in the Sn-to-Sb system. A step-wise mechanism driven by metal vacancies predominately formed at the (hk1) interfaces can be understood by comparing both crystal structures of SnSe and Sb_2_Se_3_. Their juxtaposition reveals the existence of building blocks in Sb_2_Se_3_ that are isostructural to those in SnSe (Appendix A). The SnSe-structure consists of (001) SnSe bilayers. The Sb_2_Se_3_ structure also contains such bilayers, although they are shorter along the [10] direction (3 units long) and connected to each other in a step-wise corner-sharing manner. Figure 6 depicts a plausible transformation pathway from SnSe to Sb_2_Se_3_ via an intermediate phase. The substitution of Sn with Sb through cation exchange (panel (b,c)) is followed by Sn vacancy formation (panel (d,e)). After the generation of these V_Sn_ defects (Figure 6e), the M_5_Se_6_ structure (M = Sn, Sb) consists of (Sn, Sb)Se building blocks with a length of 3 units, similar to the blocks in the Sb_2_Se_3_ structure, and they are connected through Sb–Se bonds. The Sb_2_Se_3_ structure forms upon introducing another vacancy in the metal sublattice and continued cation exchange, yielding the M_4_Se_6_ composition. All configurations shown in Figure 6 were fully relaxed to find the (local) lowest-energy minimum. The charges on the atoms were also calculated and are listed in Appendix A. For pure SnSe and Sb_2_Se_3_, the charges on the cations are 0.76*e* and 0.74*e* for Sn and Sb, respectively. These values are far from their formal valence states of +2*e* and +3*e*, indicating that the bonding is not ionic but rather covalent in nature. In the intermediate configurations in panels (b–e) of Figure 6, more charge is found on the Sb cations compared to the Sn cations (Appendix A), and Sb cations prefer to bond with undercoordinated Se atoms neighboring the V_Sn_ defects. The only exceptions are the nearest-neighboring Sb atoms in panel (c), which are forming an Sb–Sb pair with considerably less charge on the atoms. It is clear from these intermediate configurations that the substitutions and cation vacancies induce strong relaxations in the atomic lattice, which likely also lowers the migration barriers for the in- and out-diffusion of cationic species.

## 3. Summary

In summary, we have demonstrated that the cation exchange can be a facile and universal way to fabricate numerous thin-film materials with diverse compositional and structural properties without the necessity to develop individual synthetic technology for each material from SnSe: Sb and Sb_2_Se_3_ families. Our study shows that thin-film materials are more chemically mobile at moderate temperatures than previously thought. Experimental results and DFT calculations suggest that Sn-to-Sb replacement is initiated at the two-dimensional extended defect regions and can probably be explained by a vacancy-assisted cation diffusion mechanism. By varying parameters of the IE system, it becomes possible to produce numerous thin-film layers and structures, which provides an alternative strategy for the synthesis of semiconductor materials.

## 4. Methods

### 4.1. Template Film Fabrication

Continuous tin (II) selenide films were deposited onto 10 cm×10 cm square substrates using a magnetron sputtering Evovac 030 inline coating system (Angstrom Engineering, Kitchener, ON, Canada). The hot-pressed 4-inch SnSe target (4N, LOT: PLA546838312) was purchased from Plasmaterials, Inc., Livermore, CA, USA. The films were deposited at 300 °C using a target-to-substrate distance of 20 cm, plasma power of 88 W, under an argon pressure of 10^−3^ torr. The film thickness was obtained at around 700 nm. After the growth process, the films were cooled down to room temperature in the sputtering chamber for 2 h.

### 4.2. Ion Exchange

SnSe thin films were immersed into antimony chloride solutions in glycerol (2N, Fisher Scientific, Inc., Waltham, MA, USA, LOT: 2058170). The cation exchange was performed at ~210 °C under ambient conditions in open vessels (Appendix A) [14]. The concentration of SbCl_3_ (3N, Alfa Aesar, Ward Hill, MA, USA, LOT: S10F052) was changed between 11–44 mM, and the reaction time was varied within 5–22 min to tune the kinetics of the process to obtain the target Sb concentration in the films. Then, we extinguished the reaction by placing the vessel in an ice bath and using deionized water to wash off reactants. Afterward, we dried the samples in airflow at RT for 30 s and treated them in an argon atmosphere at 400 °C for 15 min to ensure the crystallinity.

### 4.3. Characterization

A Zeiss Merlin scanning electron microscope (Oerzen, Germany) equipped with the Bruker EDX-XFlash6/30 detector was employed to characterize the surface morphologies, cross-sectional views, and elemental composition of films. The SEM and EDX measurements were performed at 3 and 20 kV acceleration voltages, respectively. Quantitative analysis was performed using the PB/ZAF standard less mode.

The X-ray diffractograms were measured to reveal the structural characteristics of films using a Rigaku Ultima IV diffractometer (Neu-Isenburg, Germany) with monochromatic Cu Kα1 radiation (λ = 1.5406 Å) at 40 kV and 40 mA using a D/teX Ultra silicon strip detector. The samples were studied in a 2*θ* range of 10–70° with a scan step of 0.02°. The crystallite size was calculated by the Debye-Scherrer method [14,18]. 

The room temperature Raman spectra were collected employing a Horiba’s LabRam HR800 spectrometer (Oberursel, Germany) in the range of 50–600 cm^−1^ from a spot diameter of 100 µm using the green laser. 

### 4.4. Theoretical Calculation

The changes in the Gibbs free energies for the reactions were calculated using the Database of HSC Chemistry Ver. 6.0. by Outoukumpu Research Oy, Pori, Finland.

The Sb_2_S_3_ and Sb_2_Se_3_ thermodynamic stabilities at different pH values were estimated using the *Eh-pH* (Pourbaix) diagram [19]. The Pourbaix diagram is plotted assuming the equilibrium ion concentration ([Sb] = [S] = [Se]) in solution at the level of 10^−6^ mol L^−1^, the temperature of 298 K, and the pressure of 1 atm.

### 4.5. DFT Calculation

Density functional theory (DFT) calculations were conducted using the plane-wave VASP code [34,35]. The projector augmented wave (PAW) method [36,37] was used in combination with the generalized gradient approximation (GGA) by Perdew, Burke, and Ernzerhof (PBE) [38]. The energy cutoff of the wavefunctions and the density of the k-mesh were tested to obtain energy convergence of well within 1 meV/atom. The energy cutoff for the valence wavefunctions was set to 450 eV and the energy cutoff for the augmentation functions to 630 eV. The linear k-mesh spacing along any reciprocal lattice vector was set to less than 0.0086 Å^−1^ for metallic species and less than 0.024 Å^−1^ for Se or Cl elemental phases and Se or Cl containing compounds. The used k-meshes are listed in Appendix A. Both the dimensions of the unit cells and supercells and the atomic coordinates were allowed to fully relax to find the lowest-energy configurations. For calculation of the Cl_2_ molecule, the molecule was positioned and relaxed within a cubic box of vacuum with cube edge dimensions of 25 Å. The Bader analysis of the charges on the atoms was performed using the method of Henkelman et al [39]. Calculated lattice parameters and computed Bader charges can be found in Appendix A. 

## Figures and Tables

**Figure 1 nanomaterials-12-02898-f001:**
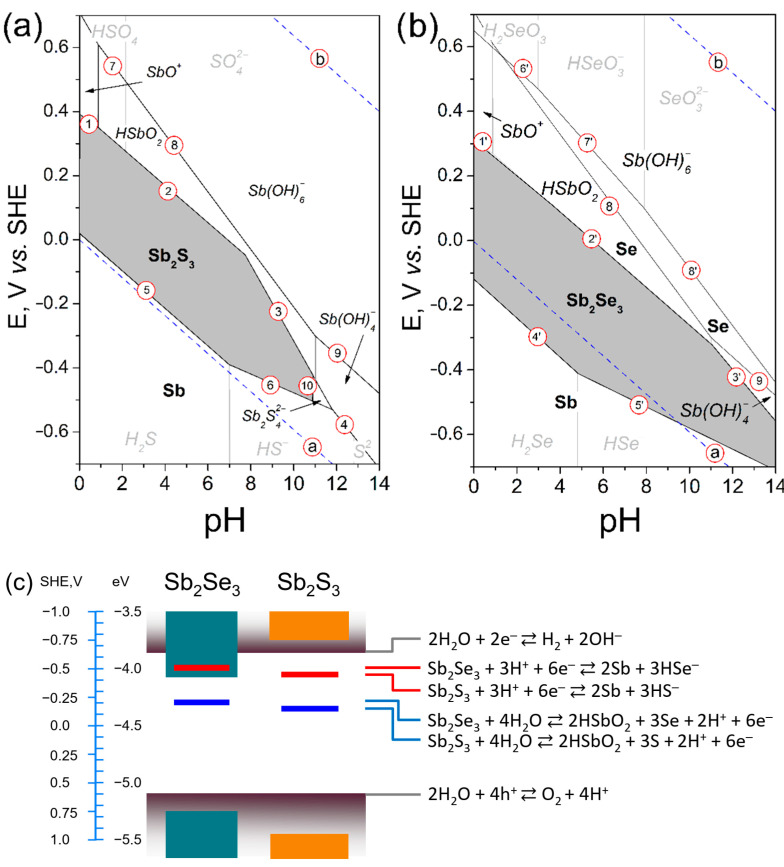
Potential-pH equilibrium diagrams for (**a**) Sb_2_S_3_-water and (**b**) Sb_2_Se_3_-water systems at 25 °C. The grey-filled areas correspond to the stability domains of Sb_2_S_3_ and Sb_2_Se_3_. The lines represent equilibria between solid Sb_2_S_3_ or Sb_2_Se_3_ and reaction products. Encircled numbers refer to the reactions listed in Appendix A. The (**a**,**b**) dashed lines correspond to the stability region of water. (**c**) Proposed decomposition scheme for Sb_2_S_3_ or Sb_2_Se_3_ in aqueous solutions.

**Figure 2 nanomaterials-12-02898-f002:**
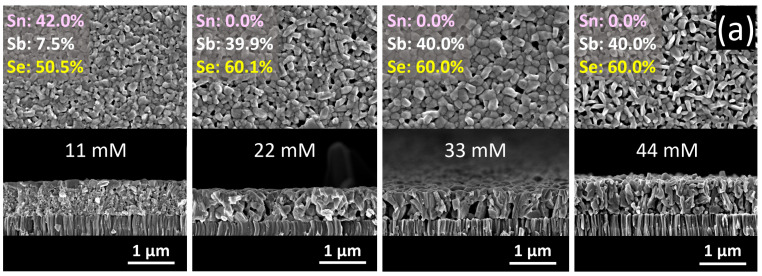
(**a**) SEM images and EDX data, (**b**) XRD patterns, and (**c**) Raman spectra of the layers that experienced cation exchange for 22 min in solutions with various concentrations of SbCl_3_.

**Figure 3 nanomaterials-12-02898-f003:**
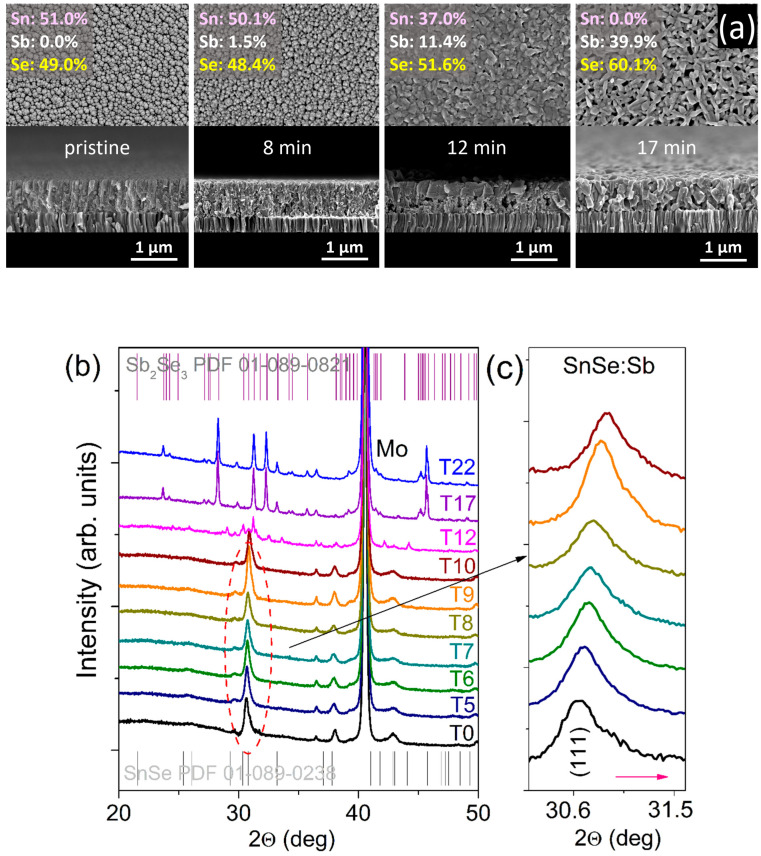
(**a**) SEM images and EDX data, (**b**) XRD patterns of the layers that experienced cation exchange in 44 mM SbCl_3_ solutions for a different time. (**c**) Magnified XRD patterns of the (111) peak of the pristine SnSe and antimony-doped SnSe layers.

**Figure 4 nanomaterials-12-02898-f004:**
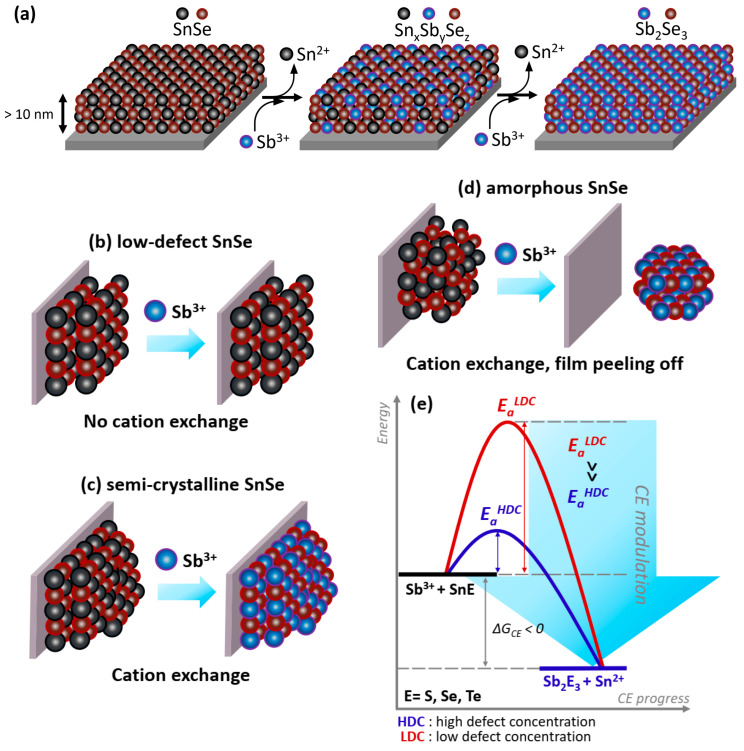
(**a**) Schematic illustration for phase transition induced by solution-assisted cation-exchange reactions. (**b**–**e**) Schematic way to control the transformation of extended solids attached to the substrate.

**Figure 5 nanomaterials-12-02898-f005:**
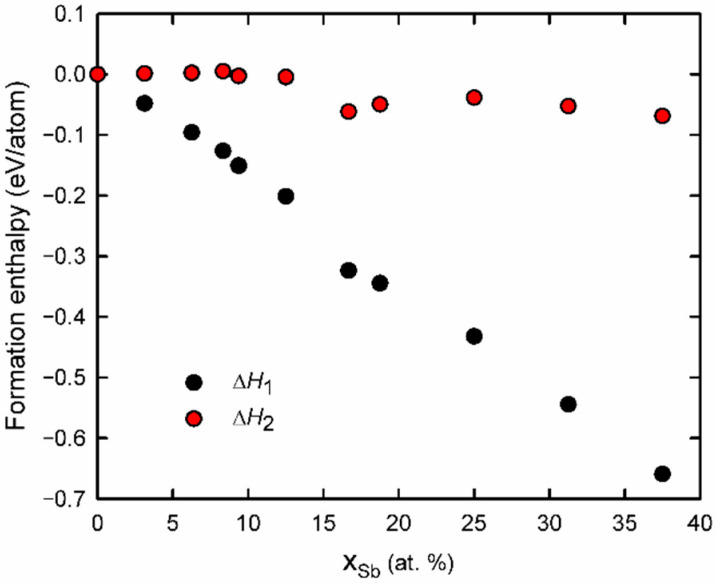
DFT-calculated formation enthalpies of Sb-doped SnSe phases plotted as a function of Sb atomic concentration, considering only potential energies. Black dots: formation enthalpy Δ*H*_1_ as defined to SnSe, Sb_2_Se_3,_ and elemental Sn using Equation (3). Red dots: formation enthalpy Δ*H*_2_ including a correction with a chemical potential shift of Sn as defined by Equation (4). The compositions and enthalpies are also listed in Appendix A.

**Figure 6 nanomaterials-12-02898-f006:**
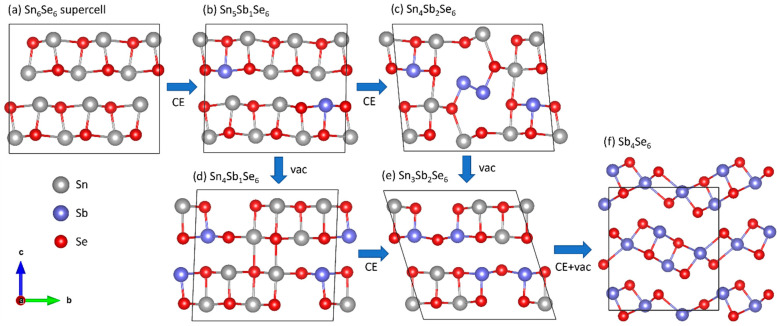
Plausible pathway of structural changes at the atomic level due to cation exchange (CE). Starting from a 1 × 3 × 1 supercell of SnSe (**a**), Sn atoms (grey) are replaced with Sb atoms (blue) while the Se atoms (red) are retained (**b**,**c**). At some point, Sn vacancies (V_Sn_) are created in the cationic sublattice (**d**,**e**), where Sb atoms preferably occupy positions around the vacancies. As a result of continued cation exchange and vacancy formation and further relaxations, the Sb_2_Se_3_ phase (**f**) is formed.

## Data Availability

Appendix A is available from the authors upon request.

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
