# Peer review of "Solution-Mediated Inversion of SnSe to Sb2Se3 Thin-Films"

_nanomaterials, 2022, doi:10.3390/nano12172898_

Round 1

Reviewer 1 Report

The Authors present a conclusive and well-analyzed study of cation exchange from SnSe to Sb2Se3. Only very minor points shall be addressed before publication.

-    1) Please explain all acronyms at first occurrence: (PCE)

-     2) Fig. 3 diagram description in the figure needs to be corrected.

Reviewer 2 Report

In this work, Polivtseva and coworkers presented an investigation on solution phase synthesis of Sb2Se3 from SnSe combining experimental and theoretical efforts. The authors showed the feasibility for fabrication of Sb2Se3 in solution phase from SnSe and they also performed thermodynamics analysis and dft based calculations to support the proposed mechanism. By carefully reading, I found the following should be addressed before the manuscript can be considered further.

l  The authors were suggested to rewrite the introduction to highlight related research results concerning the topic of the manuscript.

l  For the proposal for the mechanism of Sb2Se3, further evidences would be necessary, such as state of Sb3+ and Sn ions during the reaction, in the solution phase. This may be accomplished with in situ IR, UV, XAS, etc.

l  The resolution of Figure 1 is too low and should be revised.

l  The proposed phase diagram (Figure 1) was for aqueous solution, but the experiments were performed in glycerol at 210 degree C. Further evidence and discussions would be necessary to rational the experimental design and its relevance to the experiments.

l  It would be more interesting to highlight in Figure 1 the phase region where the synthesis of Sb2Se3 was performed.

l  Lines starting from 129, the authors used the thickness of film during reaction that doesn’t change significantly to support the proposed cation exchange mechanism, this maybe not fair considering the low solvability of both SnSe and Sb2Se3. EDX mapping would be necessary to highlight the distribution of Sb in Figures 2 and 3.

l  For the DFT proposed mechanism, it’s not clean how the lattice structure of SnSe and Sb2Se3 can be correlated considering the large discrepancies and how the Sb cations diffuse into the interlayer spacing for cation exchange and Sn species diffuse away. Please address.

l  How are the energies for structures proposed in Figure 5 and how would the results help to render the mechanism?

Reviewer 3 Report

This article deals with a very interesting transformation of SnSe thin films into Sb2Se3. The results of both experimental studies and density functional simulations are presented. The diffusion mechanism of transformation accompanied by vacancies is proposed and investigated. There are only minor remarks. It remains unclear how the chemical reaction between SnSe and SbCl3 solution leading to the formation of Sb2Se3 is finally recorded. I also did not understand well whether SnCl2 is formed during the reaction. If so, where does it disappear to, as there is no SnCl2 in Figure 2. This needs to be explained more clearly in the article.

Round 2

Reviewer 2 Report

In this revision, I saw the authors efforts on improving the manuscript. As most of my concerns were well addressed, I think the manuscript maybe publishable now. I suggest the authors to move either Figure S8 or Table S8 to the main text to easy the reading.

Author Response

We want to thank the Reviewer for his/her kind remarks and valuable suggestions on the paper. 

Following the Reviewer's suggestions, we moved the previous Figure S8 to the main text as Fig. 5.